# Potential and action mechanism of favipiravir as an antiviral against Junin virus

**Vahid Rajabali Zadeh[1], Tosin Oladipo Afowowe[1,2], Haruka Abe[1], Shuzo Urata[3], Jiro Yasuda[1,2,3]***

**1** Department of Emerging Infectious Diseases, Institute of Tropical Medicine (NEKKEN), Nagasaki University, Nagasaki, Japan, **2** Program for Nurturing Global Leaders in Tropical and Emerging Communicable Diseases, Graduate School of Biomedical Sciences, Nagasaki University, Nagasaki, Japan, **3** Department of Emerging Infectious Diseases, National Research Center for the Control and Prevention of Infectious Diseases (CCPID), Nagasaki University, Nagasaki, Japan

* j-yasuda@nagasaki-u.ac.jp

**Data Availability Statement:** All relevant data are within the manuscript and its Supporting Information files.

**Funding:** This work was supported by grants from the Japan Agency for Medical Research and

## Abstract

Favipiravir is a nucleoside analogue that inhibits the replication and transcription of a broad spectrum of RNA viruses, including pathogenic arenaviruses. In this study, we isolated a favipiravir-resistant mutant of Junin virus (JUNV), which is the causative agent of Argentine hemorrhagic fever, and analyzed the antiviral mechanism of favipiravir against JUNV. Two amino acid substitutions, N462D in the RNA-dependent RNA polymerase (RdRp) and A168T in the glycoprotein precursor GPC, were identified in the mutant. GPC-A168T substitution enhanced the efficiency of JUNV internalization, which explains the robust replication kinetics of the mutant in the virus growth analysis. Although RdRp-N462D substitution did not affect polymerase activity levels in a minigenome system, comparisons of RdRp error frequencies showed that the virus with RdRp-D462 possessed a significantly higher fidelity. Our next generation sequence (NGS) analysis showed a gradual accumulation of both mutations as we passaged the virus in presence of favipiravir. We also provided experimental evidence for the first time that favipiravir inhibited JUNV through the accumulation of transition mutations, confirming its role as a purine analogue against arenaviruses. Moreover, we showed that treatment with a combination of favipiravir and either ribavirin or remdesivir inhibited JUNV replication in a synergistic manner, blocking the generation of the drug-resistant mutant. Our findings provide new insights for the clinical management and treatment of Argentine hemorrhagic fever.

## Author summary

Development of antivirals requires cautious and extensive assessment of action mechanism as well as potential for emergence of resistant phenotype of the virus. In recent years, favipiravir has been put forward as a promising candidate for the treatment of Argentine hemorrhagic fever (AHF) caused by Junin virus (JUNV). We, therefore, aimed to provide experimental evidence on action mechanism of favipiravir to help guide its clinical use. Here we show that favipiravir causes lethal mutation that impairs virus infectivity. More

Development (AMED) (Grant No. JP20fk0108072, JP21fk0108080, JP21fk0108114 and JP21fm0208101). The funders had no role in study design, data collection and analysis, decision to publish, or preparation of the manuscript.

**Competing interests:** The authors have declared that no competing interests exist.

importantly, we demonstrate that the virus has the capability to escape favipiravir selective pressure by acquiring two amino acid substitutions on glycoprotein precursor and polymerase proteins. This observation raises concern over the use of only favipiravir in therapeutic regimens. To overcome this risk, we show that combination of favipiravir with other nucleoside analogues demonstrates a synergistic effect and suppresses the ability of JUNV to escape drug pressure. Favipiravir, ribavirin, and remdesivir have a broad spectrum of antiviral activity. Therefore, combination therapies of these drugs would be expected to have potential therapeutic effects for not only AHF but also the diseases caused by a variety of viruses, including emerging RNA viruses.

## Introduction

Argentine hemorrhagic fever (AHF) is a severe zoonotic disease that is endemic in Argentina and caused by Junin virus (JUNV). In addition to the intense clinical course of the disease, the lack of approved therapeutics and preventive countermeasures against JUNV highlights its significant threat to global public health [1–3]. Current therapeutic countermeasures against JUNV include immune plasma therapy and combinational, off-label use of ribavirin and favipiravir [3–5]. Although a live attenuated vaccine, Candid #1, was developed by the US Army Medical Research Institute of Infectious Diseases, it has been approved only for use in endemic areas due to the concerns over its genomic stability [6–8].

Favipiravir (6-fluoro-3-hydroxy-2-pyrazinecarboxamide; also known as T-705 or Avigan) is a purine analogue originally developed as an antiviral agent for influenza and subsequently reported to inhibit the replication of broad spectrum of RNA viruses [9]. Favipiravir is a prodrug that is metabolized into its active form, ribofuranosyl 5′-triphosphate (favipiravir-RTP), upon cellular uptake, and thus acts as a pseudo-nucleotide that competes with endogenous guanine and adenine nucleotides, leading to the disruption of viral replication and transcription [10, 11]. Given the extensive structural and functional similarities among RNA-dependent RNA polymerase (RdRp) of RNA viruses [12], favipiravir remains a promising countermeasure against emerging and re-emerging viral diseases caused by RNA viruses. Clinical trials showed the efficacy of favipiravir against viral hemorrhagic fever caused by the Severe Fever with Thrombocytopenia Syndrome Virus (SFTSV) and a direct correlation between favipiravir treatment and reduction in viral RNA levels [13]. However, an open-label observational study on Ebola virus showed lack of efficacy of favipiravir treatment in reducing the viral load and improving survival of the affected subjects [14]. Prior to the discovery of favipiravir, ribavirin was the only available antiviral drug effective against JUNV [15]. However, several concerns over safety and efficacy are linked with the use of ribavirin, limiting its clinical use [3]. Several pre-clinical studies investigated the inhibitory effect of favipiravir in JUNV infection [16, 17]. Favipiravir showed strong protection against lethal JUNV infection and was well tolerated at high doses. Notably, while ribavirin shows high efficacy in suppressing viral replication, the mortality of the infected animal models is only delayed or slightly reduced by ribavirin [18, 19]. Moreover, while favipiravir specifically targets the viral polymerase with minimal side effects [10], ribavirin acts through multiple mechanisms. In addition to the inhibition of viral polymerase, ribavirin targets cellular inosine monophosphate dehydrogenases and restricts intracellular GTP availability, thereby indirectly inhibiting virus replication. This explains the synergistic effect of ribavirin when used with favipiravir, which offers a combinational therapeutic approach for the clinical management of patients with AHF [20, 21].

The emergence of drug-resistant mutants abolishes the effect of antiviral drugs in the short term and therefore imposing a public health risk. To date, experimental isolation of favipiravir-resistant mutants has only been reported for chikungunya virus [22] and enterovirus 71 [23], which are positive-sense RNA viruses, as well as the influenza A virus, which contains a negative-sense, segmented RNA genome [24]. Isolation of drug-resistant mutants represents a useful approach for studying the molecular mechanisms of antiviral drugs [25]. Considering the possibility of the emergence of drug-resistant mutants and the gap in the mechanistic data on the antiviral action of favipiravir against arenaviruses [26], we attempted to isolate favipiravir-resistant JUNV. In this study, we isolated favipiravir-resistant mutants, also labeled escape mutants, with two amino acid substitutions on the viral GPC and RdRp. The analyses of the escape mutant, for the first time, provided experimental evidence that favipiravir primarily acts against JUNV by inducing transition mutations. Here, we showed that the selective pressure of favipiravir promoted the emergence of the JUNV variant with a higher replication fidelity and, therefore, lower susceptibility to favipiravir. We also showed that the treatment with combination of favipiravir with either ribavirin or remdesivir inhibited JUNV replication in a synergistic manner. It may be useful as an effective treatment without the risk of the emergence of drug-resistant mutants.

## Results

### Isolation of favipiravir-resistant mutants

Candid #1 is the only live-attenuated vaccine strain (Candid #1) of JUNV, which is available for clinical use and can be handled in BSL-2 laboratory. Therefore, we used Candid #1 as a model strain of JUNV in this study. To isolate the favipiravir-resistant mutant, Candid #1 was serially passaged in 293T cells in the presence of favipiravir. We chose this cell line, since we observed a robust replication kinetics of Candid #1 as compared to other cell lines that we tested. First, to identify the optimal selective pressure of favipiravir on JUNV replication, the $IC_{50}$ value was determined by a dose-response experiment. 293T cells were infected with Candid #1 virus at a multiplicity of infection (MOI) of 0.1, in the presence of a series of favipiravir dilutions or DMSO. Quantification of viral titers at 48 hours post infection (hpi), showed favipiravir $IC_{50}$ to be 4.9 μM with 95% confidence interval (CI) of 4.5 μM to 5.4 μM. This value is comparable with that of a previous study reporting favipiravir $IC_{50}$ for JUNV in Vero cells [27]. The cytotoxicity assay showed that favipiravir lacked toxic effects on cell viability at the specified concentrations (**S1 Fig**). We selected 5 μM of favipiravir as an optimal selective pressure. First, the cells were infected with the JUNV Candid #1 strain (MOI: 0.01). After adsorption, medium containing either favipiravir or DMSO was added, and cells were then incubated. We thought to have an additional, no-passage control using a fresh virus stock that was used to compare with DMSO-passaged controls, since it has been reported that the repeated passaging of some RNA viruses in cell culture is associated with the production of viral particles containing defective viral genome [28, 29]. Next, viral titers were measured using plaque assay after every passage. In the first two initial passages, P0 and P1, JUNV titers showed a 81.7 and 105-fold reduction, respectively (**Fig 1A**). To increase the selective pressure, favipiravir concentration was increased to 20 μM (approximately $IC_{90}$) for subsequent passages. The increase of favipiravir concentration was associated with a reduction in titer similar to that detected using 5-μM favipiravir, indicating that a resistant population was beginning to emerge. As shown in **Fig 1A**, at passage 11 (P11), viral titers were similar to those of no-drug controls, suggesting that a dominant proportion of the viral population was resistant to favipiravir. Further passaging (P12 and P13) in the presence of favipiravir did not affect viral titers. To measure the reduction in susceptibility of the P11 virus to favipiravir, a dose-response assay

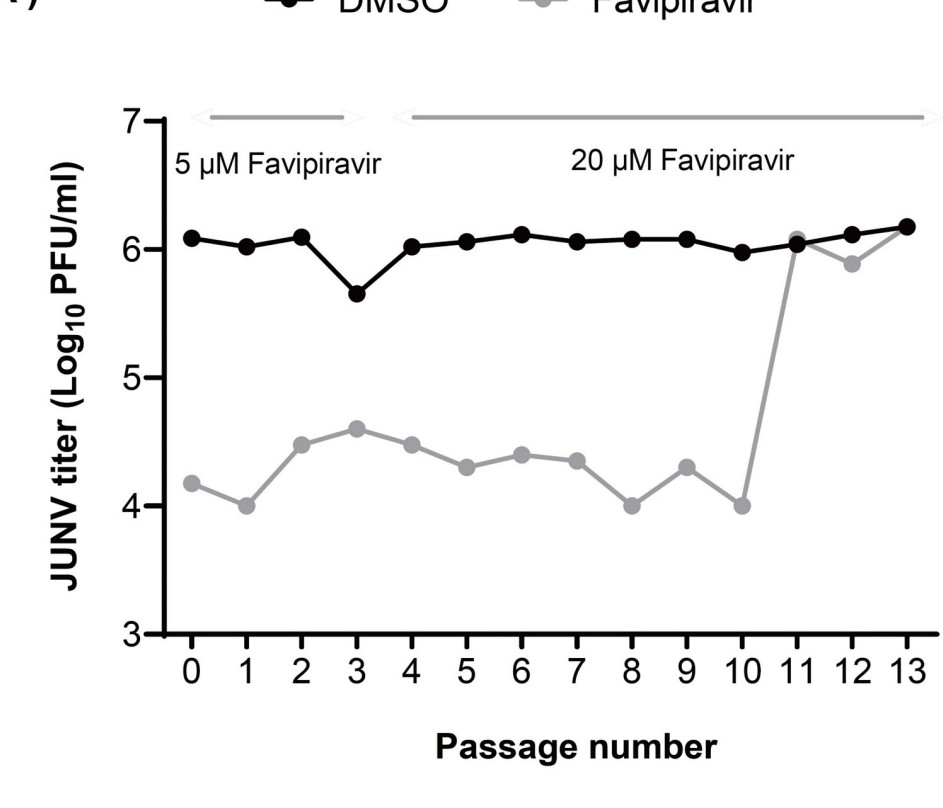

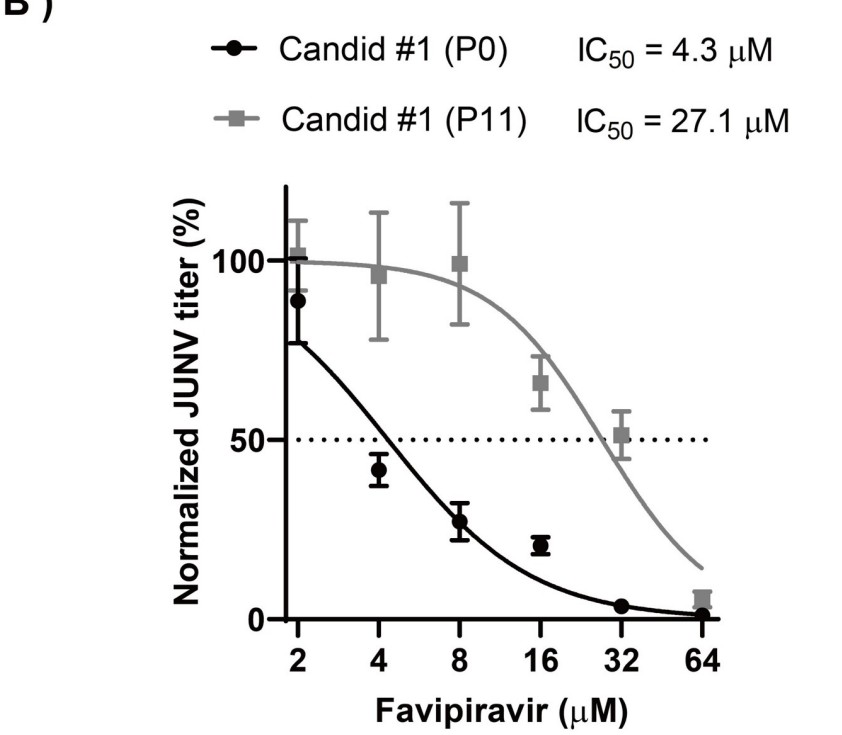

**Fig 1. Emergence of favipiravir resistant JUNV by sequential passaging in 293T cells.** (A) Serial passage of JUNV in presence of moderate favipiravir concentrations. JUNV was passaged in 293T cells (MOI: 0.01) in presence of 5 μM for

the first three passages and 20 μM favipiravir for the remaining passages. After a total of 11 passages, a resistant JUNV population emerged (gray). Control passages were performed in parallel using DMSO (black), n = 1. (B) Favipiravir dose-response analysis. 293T cells were infected with JUNV Candid #1 parental (P0) or passage 11 (P11) viral populations (MOI: 0.1), After adsorption, media containing indicated concentrations of favipiravir was added to the cells. At 48 hpi., viral titers were measured by plaque assay. Error bars indicate ±SD; three independent experiments in duplicate (n = 6) were performed; nonlinear regression analysis was applied.

was performed. As shown in **Fig 1B**, $IC_{50}$ values of P0 and P11 were 4.3 μM (95% CI = 3.8 to 4.9) and 27.1 μM (95% CI = 23.1 to 31.6), respectively, indicating a significant increase in $IC_{50}$ value (6.3-fold) and the emergence of favipiravir-resistant Candid #1-mutant (Candid #1-res).

## Identification of the mutations in the favipiravir-resistant JUNV mutant

To identify the mutations that confer resistance to favipiravir, four clones were isolated from P11 by plaque assay, as described in the Materials and Methods section. Viral RNA from each clone was extracted, and the sequences of all open reading frames (ORFs) were determined. Three nucleotide substitutions, two in the RdRp coding region (1384 A to G, 3669 A to G) and one in the GPC coding region (502 G to A), were identified in P11 mutants **(Fig 2A)**. In the RdRp region, the A to G substitution at 3669 generated a synonymous mutation, whereas the A to G substitution at 1384 led to N462D amino acid modification in the PA-like domain of RdRp [30, 31]. In the GPC region, G to A substitution at 502 caused an A168T amino acid substitution within the GP1 subunit. No mutations were observed in Z or NP genes. Next, to examine the proportion of the mutations, N462D in RdRP and A168T in GPC, next-generation sequencing was performed with PCR products containing the mutation site at the time-point of P2, P4, P6, P8, P10, and P11. As shown in Fig 2B, for RdRP, more than 80% of viruses possessed the original N462 residue from P2 to P8, then the D462 mutant suddenly increased its proportion to 57.3% at P10. Whereas the proportion of viruses possessing the original A168 residue in GPC was dramatically reduced with the passage number: P2, 91.1%; P4, 49.1%; P6, 21.1%; P8, 22.1%. The proportion of the original A168 virus was only 0.79% at P10. Finally, complete replacements of both substitutions were observed at P11. These results suggested that the Candid #1 JUNV obtained the GPC A168T mutation in the continuous pressure of Favipiravir and the RdRP N462D mutation to acquire Favipiravir resistance **(Fig 2B)**.

## Growth kinetics of favipiravir-resistant JUNV mutant

First, we compared the growth kinetics of Candid #1-res (P11) to parental Candid #1 (P0) in the absence of favipiravir. After virus adsorption on ice to synchronize the infection as described in the Materials and Methods, 293T cells infected with either Candid #1 or Candid #1-res were incubated and culture supernatants were collected at 8, 12, 24, and 28 hpi. Viral titers in the culture supernatants were determined by plaque assay. Efficient replication of Candid #1-res was observed at 8 hpi, which was much earlier than that of parental Candid #1 (**Fig 3**). However, there was no significant difference between the titers of both viruses at 28 hpi, indicating that Candid #1-res exhibited rapid growth. No differences in plaque morphologies were observed.

## High-fidelity replication of favipiravir-resistant JUNV

To understand the mechanism by which JUNV acquired resistance against favipiravir, the mutation frequency of Candid #1 and Candid #1-res in the presence of favipiravir was assessed. Following an established approach that has been shown to be sensitive enough to distinguish fidelity differences by targeting regions as few as 100 base pairs [32], first 293T cells

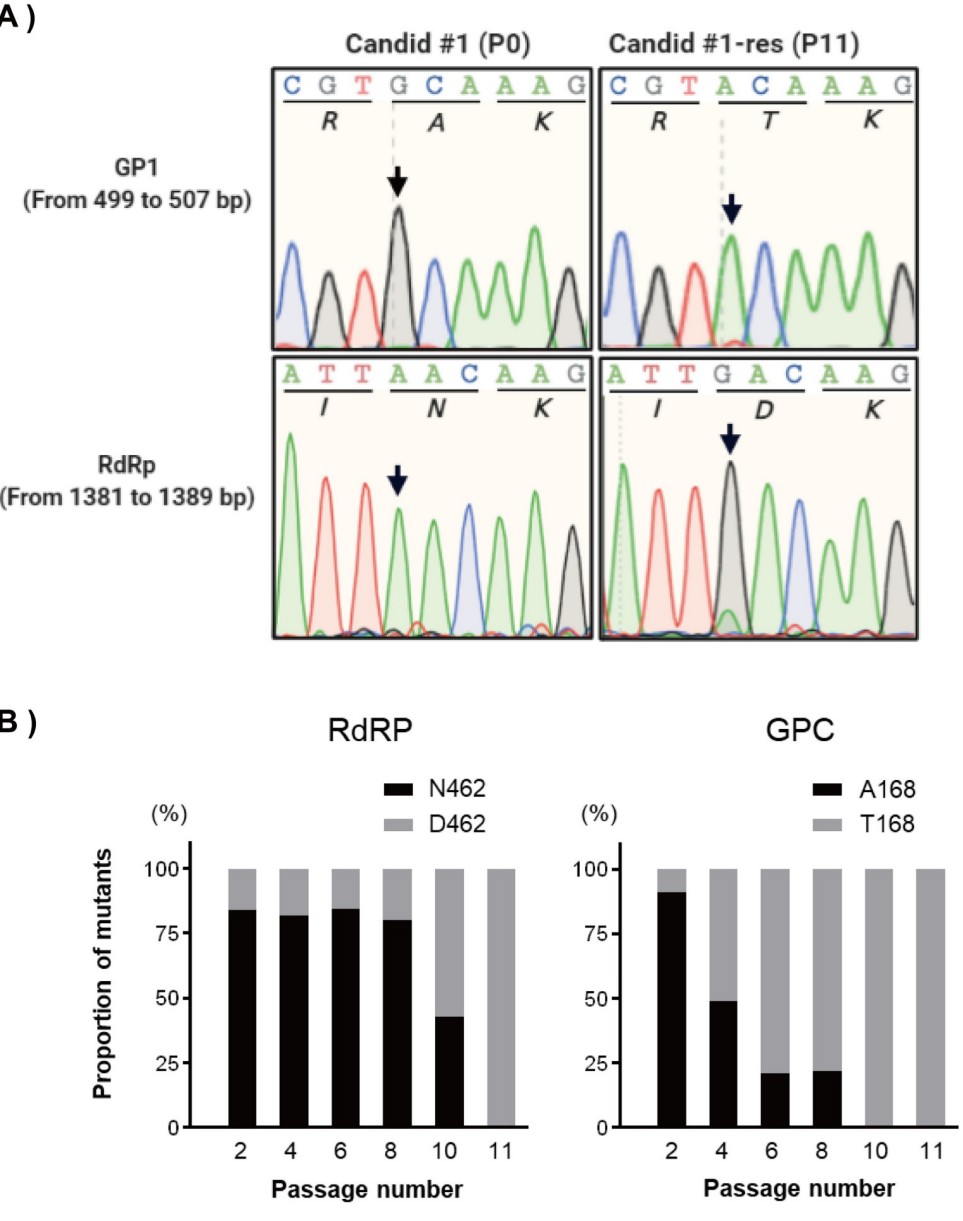

**Fig 2. Nucleic acid substitutions in RdRp and GPC open reading frames.** (A) Representative chromatograms of mutations of favipiravir resistant JUNV Candid #1, passage 11 (P11) are shown in comparison to the parental virus population (P0). Amino acid residues are shown below each codon. Arrows indicate mutation locations. (B) The proportions of RdRp-N462D (left panel) and GPC-A168T (right panel) mutations in every other passage.

were infected with Candid #1 or Candid #1-res (MOI: 0.1) and incubated in the presence of 20 μM favipiravir. After 48 h, the culture supernatants were collected. Next, we cloned the same region of the NP gene from each virus and performed clonal sequencing. We chose nucleoprotein (NP) gene because it's highly prone to mutation during viral replication [33] therefore, allowing us to estimate the baseline error rate of each virus. As shown in **Fig 4**, the mutational analysis of a total of 24,300 nucleotides for the parental Candid #1 and 27,450 nucleotides for the Candid #1-res showed that the resistant mutant had acquired 1.82 mutations per 10,000 nucleotides as compared to the 8.64 mutations of the parental population

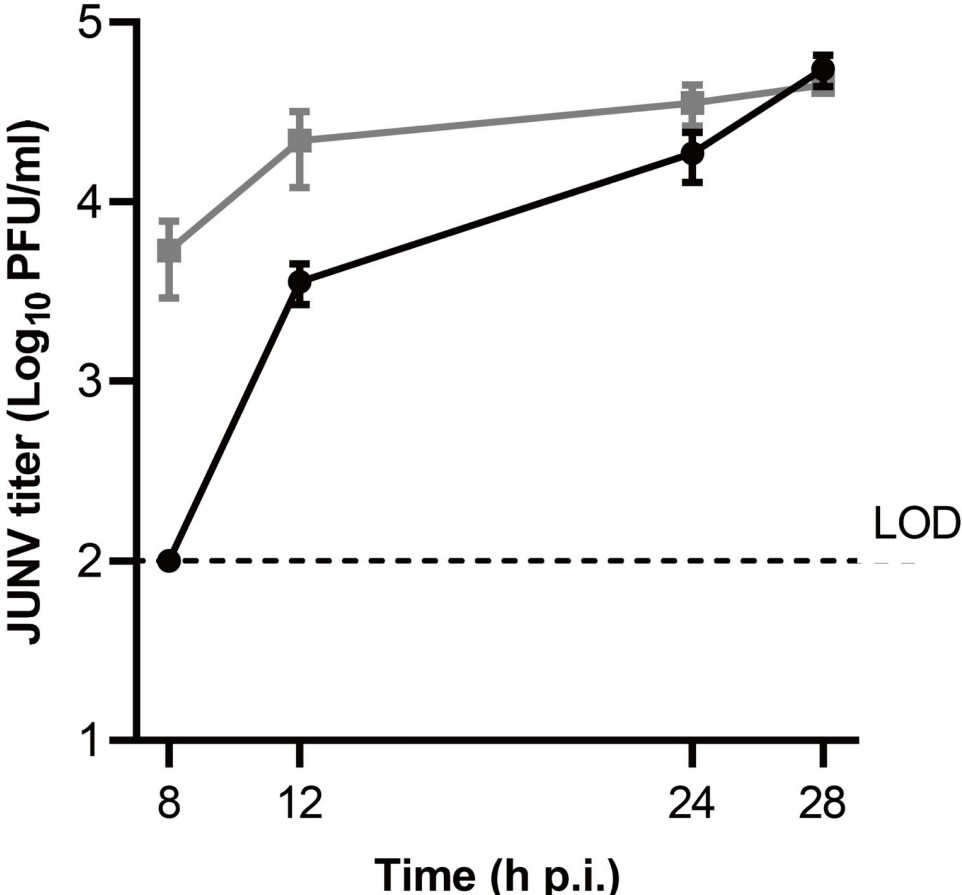

**Fig 3. JUNV Candid #1-mutant virus replication begins at an earlier time point.** To determine the one-step growth kinetics of favipiravir resistant JUNV, 293T cells were infected with either Candid #1 or Candid #1-mutant (MOI: 0.1). In order to synchronize infection, cells were incubated on ice for 30 minutes. Cells were then washed with PBS (−), and pre-warmed DMEM containing 10% FBS was added. Supernatant was collected at the indicated time points. Titers were determined by plaque assay. Error bars indicate ±SD, three independent experiments in duplicates (n = 6) were performed. Statistical significance was determined by 2-way ANOVA test ($P < 0.0001$); LOD, limit of detection (100 PFU/ml).

(4.7-fold lower. $P = 0.0053$, Mann–Whitney $U$ test), indicating higher fidelity of the Candid #1-res virus. Similarly, mutation frequencies of Candid #1-res in DMSO-treated controls were slightly lower (3.8-fold), although the data lacked statistical significance ($P = 0.242$, Mann–Whitney $U$ test). A 3 to 5-fold increase in mutation rate is known to cause lethal mutagenesis [34, 35]. Therefore, we estimate the frequency of favipiravir-induced mutations observed in the parental Candid #1 to be beyond the tolerable threshold of error-catastrophe for Junin virus. This is comparable to those reported for other viruses, including influenza, Rift Valley fever, Zika, and murine norovirus treated with favipiravir [11, 36–38]. These data suggest that mutagenesis with lethal consequences is the primary mechanism of favipiravir antiviral action against JUNV.

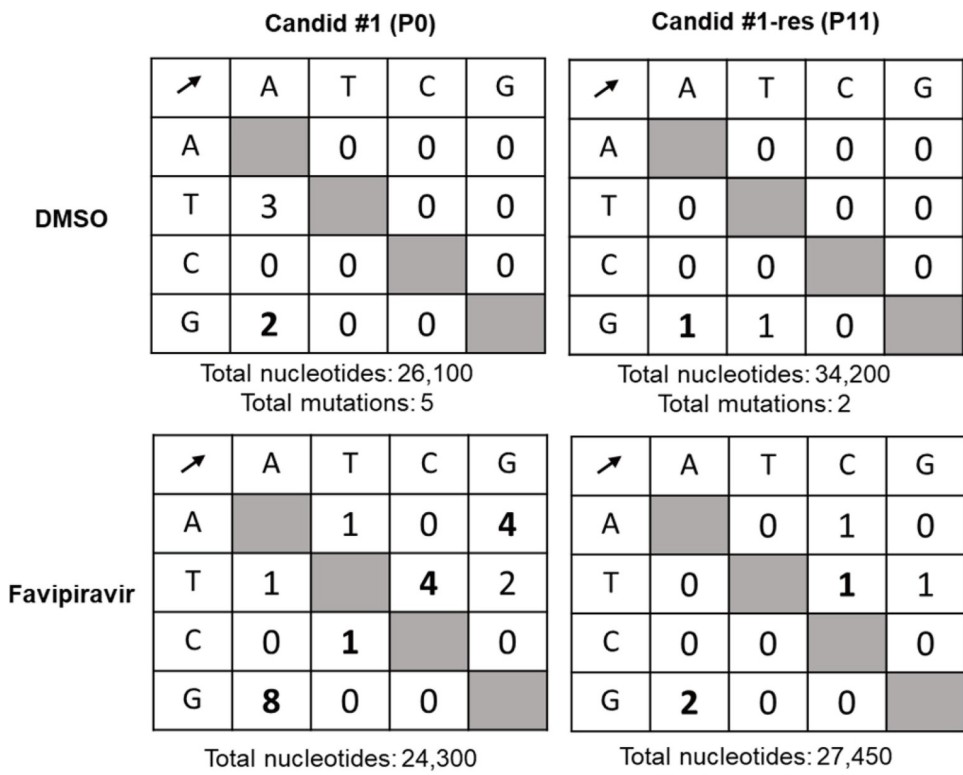

**Fig 4. Mutational frequencies of virus populations in presence of 20 μM favipiravir or DMSO control.** Clonal sequencing targeting 450 bp of nucleoprotein (NP) gene was performed (as described in materials and methods) to estimate the frequency of mutations in each virion population. Nucleotide polymorphisms were counted for Candid #1 and Candid #1-res viruses to estimate the proportions of transition (shown in bold) and transversion substitutions in presence of 20 μM favipiravir.

Next, we categorized the substitutions to determine the proportions of transitions versus transversions. The most common mutations in the presence of favipiravir were G to A, followed by T and C mutations, accounting for 80% of all substitutions in favipiravir-treated viruses, indicating that the mutation profile induced by favipiravir is biased towards transitions (**Fig 4**). This is in agreement with studies on other viruses treated with favipiravir [11, 36, 39, 40]. Consistent with the literature, supplementation of purines, but not pyrimidines, reversed the antiviral activity of favipiravir (**S2 Fig**) [41], reaffirming the role of favipiravir as a purine analogue that competes with adenosine and guanosine during nucleotide incorporation. This, in turn, explains the error bias observed in the spectrum of mutations induced by favipiravir.

## Reduced specific infectivity of favipiravir-resistant JUNV

To further evaluate the mutagenic effect of favipiravir on Candid #1 and Candid #1-res viruses, their respective specific infectivity (defined as the ratio of infectious virions to the encapsidated genome copy number), when exposed to increasing concentrations of favipiravir, were compared. 293T cells were infected with Candid #1 or Candid #1-res (MOI: 0.01), and then treated with serial dilutions of favipiravir or DMSO. Specific infectivity of Candid #1 at 48 hpi was 0.85 and 0.39 $\log_{10}$ plaque-forming units (PFU) per mL/$\log_{10}$ RNA copies per mL for 2 μM and 64 μM favipiravir, respectively, and showed a concentration-dependent reduction (**Fig 5**).

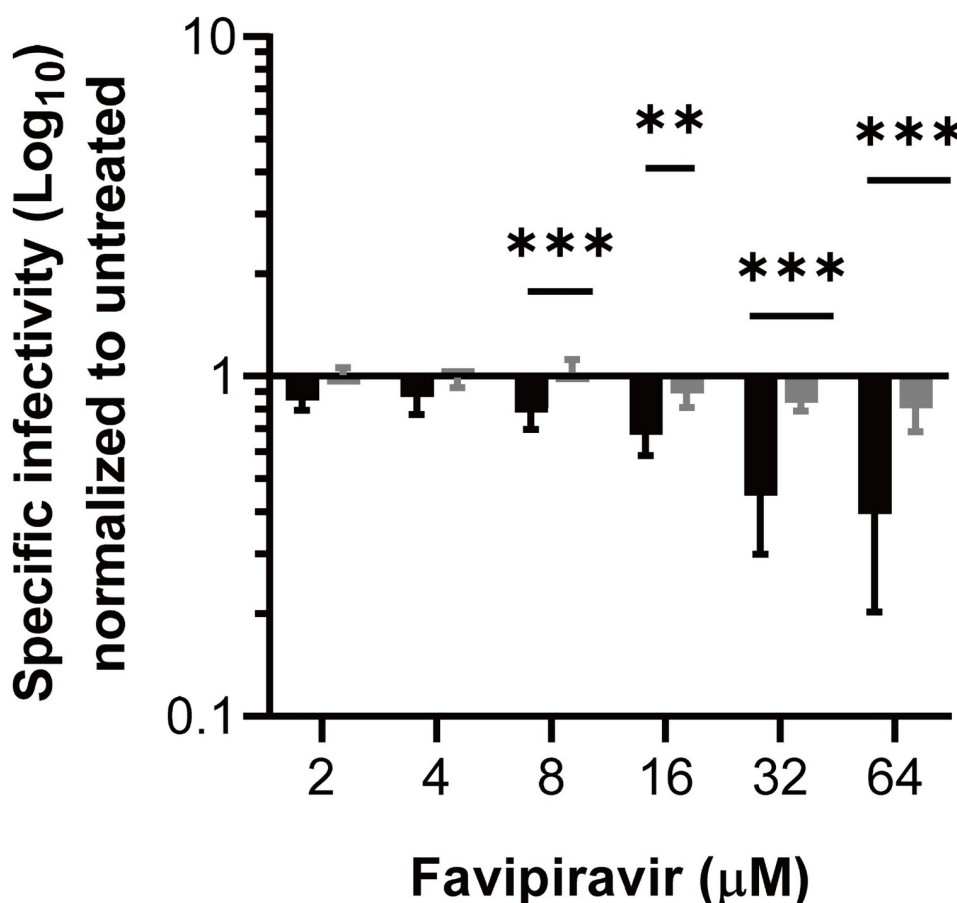

**Fig 5. Specific infectivity of Candid #1 and Candid #1-res viruses.** 293T cells were infected with each virus (MOI: 0.01) and treated with different concentrations of favipiravir or DMSO. Specific infectivity values at 48 hpi, were calculated using the ratio of infectious particles ($Log_{10}$ PFU/mL) to the RNA copy numbers ($Log_{10}$ copies/mL). Values were normalized to DMSO-treated controls. Error bars indicate ±SD; two independent experiments in four biological (n = 8) were performed. Statistical significance was determined by 2-way ANOVA test (** indicates $P < 0.01$ and *** indicates $P < 0.001$).

This suggests that increasing concentrations of favipiravir caused the accumulation of lethal mutations that lead to the loss of JUNV infectivity [36, 42]. In contrast, specific infectivity of Candid #1-res virus was only slightly affected from 1.009 to 0.80 $log_{10}$ PFU per mL/$log_{10}$ RNA copies per mL, when exposed to 2-μM and 64-μM favipiravir, respectively. The significant reduction in specific infectivity of Candid #1 compared to that of the Candid #1-res virus ($P < 0.0001$, two-way ANOVA test) suggests that the resistant mutant is less susceptible to the mutagenic effect of favipiravir. Notably, we did not observe any difference in viral RNA copy numbers between Candid #1 and Candid #1-res variants across any concentration (up to 64 μM) of favipiravir treatment, further indicating the presence of genomic RNA that are not capable of producing infectious virus. A reduction in specific infectivity is the hallmark of

error-catastrophe-mediated extinction [43, 44] and thus, these data indicate that the lower susceptibility of Candid #1-res virus to favipiravir is mediated by its higher replication fidelity.

## Enhancement of JUNV growth by GPC-A168T substitution

To investigate the functional impact of GPC A168T substitution on virus entry, we compared the internalization dynamics of favipiravir-susceptible (P0) and resistant variants (P11) using a pseudotyped vesicular stomatitis virus (VSV) system in 293T cells. To normalize the number of viral particles used for infection, real time qPCR targeting the VSV-M gene was performed (**S3 Fig**). A confluent monolayer of cells was infected with the pseudotyped VSV bearing GPC-A168 or GPC-T168, Candid#1pv-A168, or Candid#1pv-T168, which have equivalent copies of the VSV genome. To allow synchronized virus entry, cells were first incubated at 4˚C for 30 min and subsequently transferred to 37˚C for further incubation. Measurements of luciferase signal at 8, 12, 16, and 20 hpi showed a significant difference at 12 and 16 hpi, with Candid#1pv-T168 having more robust entry kinetics compared to Candid#1pv-A168 (**Fig 6A**). To rule out the possibility that the enhanced virus internalization is the result of more efficient virus attachment, we determined the genome copy number of attached viral particles. The result showed that there was no significant (two-way ANOVA test) difference in virus attachment efficiencies (**Fig 6B**). We then examined the intracellular levels of the VSV-M protein as a marker of fusion efficiency. Using a similar experimental setup, 293T cells were infected with an equal number of viral particles (input virus). After 8 h, the cells were lysed, and samples were prepared for the detection of VSV-M protein using western blotting (**Fig 6C**). Candid#1pv-T168 showed 12 times more intracellular M protein expression levels than Candid#1pv-A168, despite similar levels of input virus (**Fig 6D**), indicating that the viral genome was released into the cytoplasm more efficiently, leading to more rapid and elevated VSV-M protein expression. These data suggest that Candid#1pv-T168 is more efficient in the entry and/or fusion processes, leading to an altered viral life cycle.

## Effect of RdRp-N462D substitution on RNA polymerase activity

Next, we investigated whether N462D substitution affects RdRp activity in Candid #1. A minigenome (MG) system based on the S segment of Candid #1 virus was constructed, and an MG assay in absence of favipiravir was performed. To ensure that both RdRp-N462 and -D462 plasmids had comparable expression levels, both proteins were tagged with a FLAG peptide and similar expression levels were confirmed by western blot analysis (**Fig 7A and 7B**). 293T cells were transfected in 24-well plates with the plasmids for either NP, MG, RdRp-N462, RdRp-D462, or empty vector. Luciferase signals (normalized to the internal control) were measured at 24 hours post-transfection (hpt). The results are expressed as relative induction rates. We observed that N462D substitution had no significant effect on RNA polymerase activity in the absence of favipiravir (**Fig 7C**). To examine the sensitivity of RdRp-D462 to favipiravir, the activity of both polymerases exposed to different concentrations of the drug using a minigenome system was tested. No significant difference in favipiravir dose-response in RdRp-N462 ($IC_{50}$ = 245.6–304.3 μM [95% CI]) and RdRp-D462 ($IC_{50}$ = 289.8–362.1 μM [95% CI]) was observed, although RdRp-D462 showed a slightly increased resistance to favipiravir compared with that of RdRp-N462 (**Fig 7D**). No cytotoxicity was observed in this assay (**Fig 7E**). While lower concentrations of favipiravir only slightly inhibited MG expression, we observed that the inhibition of luciferase signal at higher concentrations ($\geq$200 μM) is associated with the reduction of mRNA levels, suggesting favipiravir mainly inhibits our minigenome system via a chain termination-mediated mechanism. (**Fig 7D and 7F**).

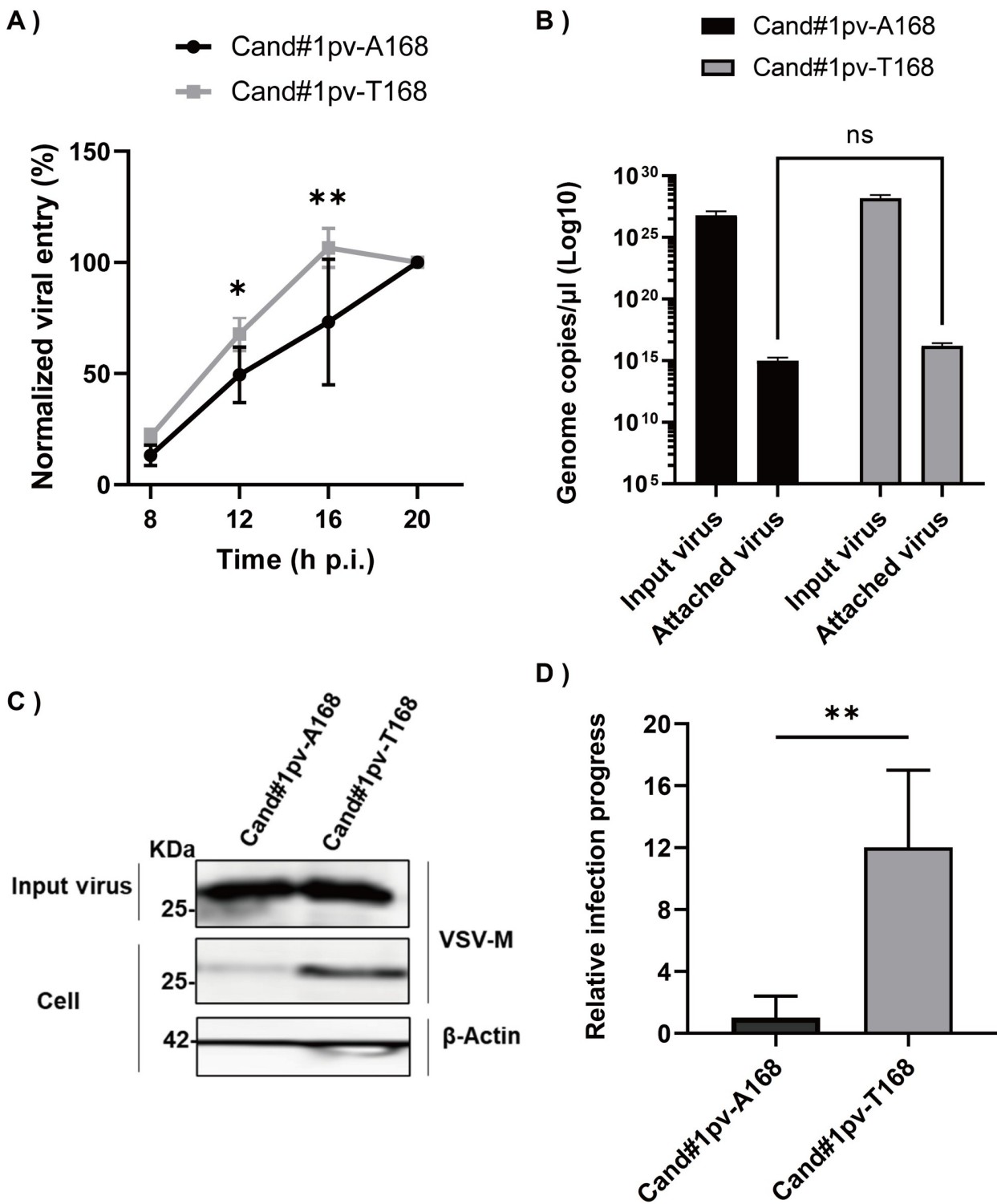

**Fig 6. GPC-A168T substitution enhances viral entry dynamics.** (A) 293T cells were infected with Candid#1pv-A168 or Candid#1pv-T168 virus at 4˚C to allow synchronized attachment. Unbound viral particles were removed and internalization was initiated by transferring the cells to 37˚C. Luciferase signal was measured at indicated time points. Signal at 20 hours was considered 100%. (B) Attached viral particles were quantified and normalized to the inoculum. (C) Western blot analysis (anti-VSV-M in the upper panel for the detection of the VSV-M protein, and Anti-β-Actin as loading control) to assess fusion efficiency of pseudotyped viruses. (D) Expression levels of intracellular M protein was normalized to the input virus. Quantified results of two independent experiments (n = 6) are shown. The bar indicates ±SD. Statistical significance was determined using 2-way ANOVA, (* indicates $P < 0.05$ and *** indicates $P < 0.001$).

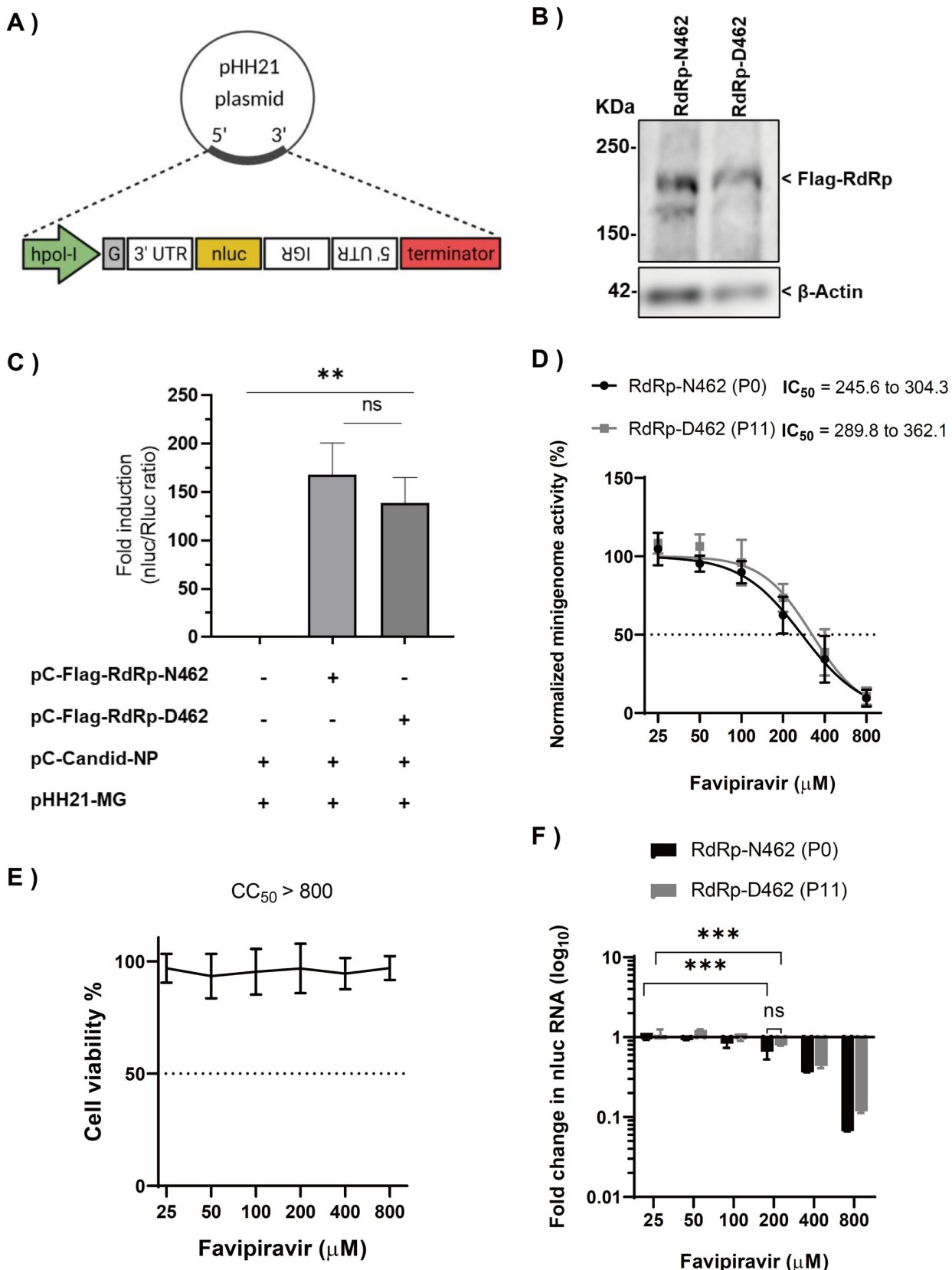

**Fig 7. Effect of RdRp-N462D substitution on polymerase activity.** (A) Schematic representation of the minigenome (MG) plasmid constructs with a nluc reporter (B) Western blot analysis (anti-Flag; upper panel for the detection RdRp-N462 or RdRp-D462 protein. Anti-β-Actin; loading control) to ensure equal expression levels (C) Polymerase activities measured using the MG system in 293T cells at 48 hpt. Results are expressed by the ratio of nano luciferase to renilla luciferase activity (internal control). Quantified results of two independent experiments (n = 6) are shown. The bar indicates ±SD. Statistical significance was determined using *t*-tests (** indicates $P < 0.01$. ns indicates not significant). (D) Sensitivity of RdRp-N462 and RdRp-D462 polymerases were compared using the MG system. Quantified results of two independent experiments (n = 6) are shown. (E) Cell viability assay as described in materials and methods. (F) Quantification of nLuc mRNA from a minigenome assay by qPCR. ΔΔCT was calculated using *GAPDH* as described in the materials and methods. Fold change in mRNA was normalized to DMSO-treated controls. Error bars indicate ±SD; two independent experiments in duplicates (n = 4) were performed. Statistical significance was determined using a 2-way ANOVA test (ns indicates not significant, ** indicates $P < 0.01$ and *** indicates $P < 0.001$).

## Combinational inhibitory effect of favipiravir with either ribavirin or remdesivir on JUNV growth

Combination antiviral therapy is a promising approach to minimize the risk of the emergence of drug resistance and enhance the antiviral effect. Therefore, we investigated the inhibitory effect of favipiravir in combination with ribavirin and remdesivir. As shown in **Fig 8,** the anti-JUNV effect of favipiravir was significantly higher when combined with ribavirin (ZIP synergy score: 14.02) or remdesivir (ZIP synergy score: 15.82) without any significant antagonistic effect. No cytotoxicity was associated with any of the tested drug combinations. Despite our attempt to isolate a resistant variant to combinational treatments, no resistant variant was generated even after 15 passages (**S4 Fig**).

## Discussion

In this study, we attempted to understand the mechanism of antiviral action of favipiravir against JUNV by isolating resistant variants. In our approach, lower concentrations of favipiravir for three initial passages followed by higher concentrations for the remaining passages were used. This gradual increase in selective pressure allowed adequate levels of replication to continue without a sudden exposure of viral sub-populations to the lethal concentrations of favipiravir [45], thus enabling us to successfully maintain and isolate the resistant population.

The arenavirus RdRp consists of three domains: an N-terminal PA-like domain with endonuclease activity, a polymerase region possessing the active site, and a PB2-like domain. In contrast to previous studies showing that RNA viruses developed resistance to favipiravir through mutations in the conserved catalytic domain of viral RdRp [22–24], we identified an RdRp-N462D substitution within the PA-like domain in favipiravir-resistant variants. A recent study, which resolved the structure of arenavirus polymerase protein with a near-atomic resolution showed that residue 462 of new world arenaviruses belongs to the core lobe region of the PA-like domain, which is involved in the stabilization of the polymerase active site. However, the precise interactions of the 462 residue are yet to be clarified [30]. Our assessment of the functional impact of N462D substitution using an MG system showed only a slight, statistically non-significant ($P = 0.125$) reduction in reporter activity without any major impact on polymerase function (**Fig 7C**). However, comparisons of mutation frequencies of Candid #1 and Candid #1-res virus revealed a significant reduction in polymerase error number in the case of RdRp-D642, suggesting an important role of this residue in polymerase fidelity (**Fig 4**). The leading hypothesis on the mechanism of the favipiravir resistance observed in this study is that the N462D substitution enhances the selectivity of RdRp for the correct nucleoside triphosphates during replication and transcription, resulting in lower favipiravir incorporation, as described for other mutagen-resistant RNA viruses [46, 47]. Further analysis of binding affinities will clarify the precise mechanism of D462 resistance to favipiravir. Notably, the higher fidelity of Candid #1 mutant virus correlated with resistance to favipiravir, and the

A )

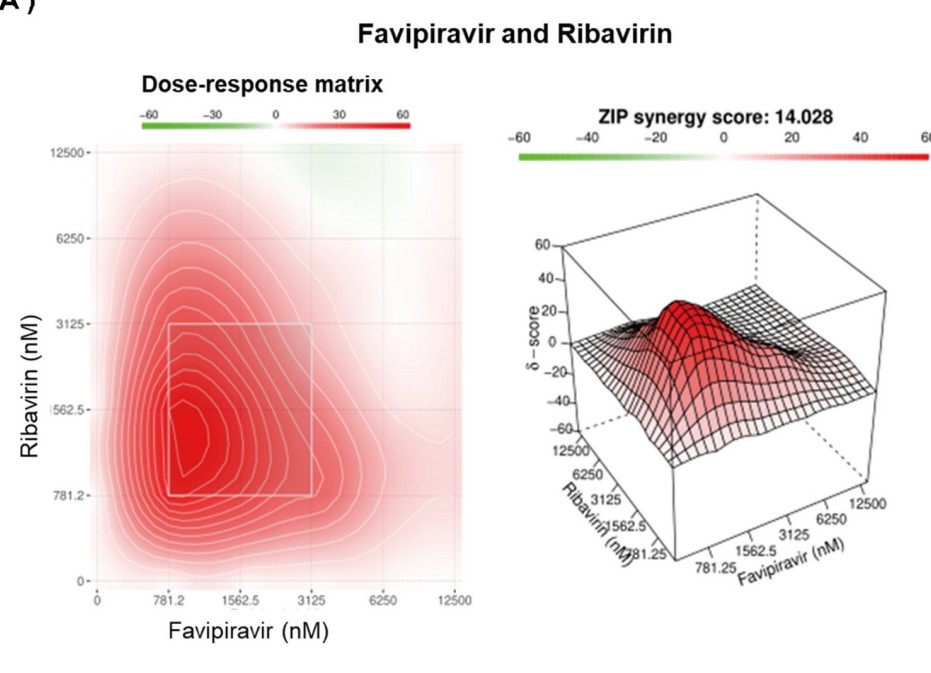

B )

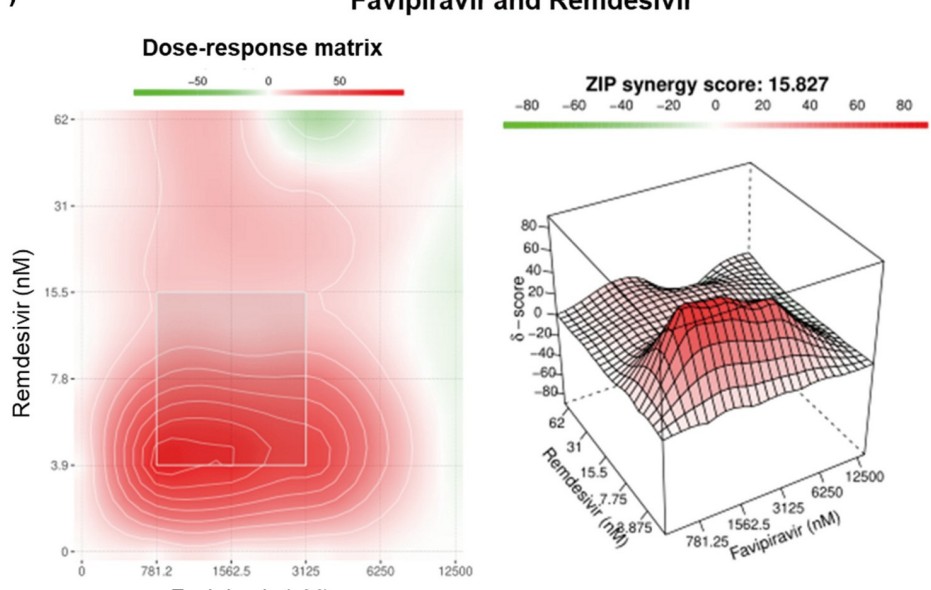

**Fig 8. Combination inhibitory effect of favipiravir and ribavirin or remdesivir on JUNV.** 293T were infected with JUNV (MOI: 0.1) and subsequently treated with a 6 × 6 drug combination matrix of favipiravir + ribavirin (A) or favipiravir + remdesivir (B) Dose-response matrix and synergy heat map are presented. Colored bar indicates the strength of synergy (δ-score); less than −10 is likely to be antagonistic, −10 to 10 suggests an additive drug interaction, and larger than 10 indicates a synergistic effect. Data are means of two independent experiments in duplicates (n = 4).

virus remained susceptible to higher concentrations of the drug, indicating that RdRp-D462 does not tolerate the chain termination activity of favipiravir, as was demonstrated in this study with the MG system (**Fig 7D and 7F**). Accordingly, the Candid #1-res virus remained

susceptible to other purine analogues, ribavirin which has both mutagenic and non-mutagenic mechanisms of action and remdesivir which is a non-mutagenic replication inhibitor, further supporting our conclusion of higher fidelity of RdRp-D462 (**S5 Fig**) [10, 41, 48, 49]. To date, with the exception of influenza virus [46], other RNA viruses with high replication fidelity possess a positive-sense, non-segmented genome [47, 50]. To the best of our knowledge, this is the first report on the isolation of a high-replication fidelity phenotype amongst hemorrhagic fever viruses. Studies have demonstrated that higher fidelity of replication affects the genetic heterogeneity of viral sub-populations, imposing a fitness cost *in vivo* [46, 51, 52]. Hence, there remains a need to further investigate the virulence and pathological characteristics of JUNV with high-fidelity replication, which was isolated in this study.

The other mutation (A168T) identified in this study was found to be within the GP1 subunit of the glycoprotein complex (GPC). Arenavirus GPC is a precursor protein that forms a trimer of stable signal peptides, GP1, and GP2 subunits, upon maturation by cellular enzymes. During virus entry, GP1 and GP2 are responsible for the recognition of receptors and the fusion with endosome membranes, respectively [53]. The GPC-168 residue has been extensively studied as it is known to partially contribute to the attenuation mechanism of Candid #1 vaccine strain. Existing evidence shows that T168 residue restores an N-link glycosylation motif within GP1 thereby enhancing GPC trafficking from endoplasmic reticulum to the cell surface [54–57]. Here we take a step further and show the importance of this residue in virus infectivity. While A168T substitution led to more efficient viral entry, no impact on attachment of pseudotype viral particles to the target cells was observed (**Fig 6**), suggesting that the functional effect of the A168T substitution is on the post-attachment step of JUNV virus entry. Consistent with this, growth kinetics of the Candid #1-res virus represented more robust replication at earlier time points (**Fig 3**). As we traced the emergence of both mutations in different passages, we observed that the GPC-A168T substitution occurred earlier and in higher proportions as compared to RdRp-N462. Considering its role in virus entry it is possible that GPC-T168 contributes to favipiravir resistance by increasing virus fitness. More recently, a novel mechanism of drug resistance mediated by an altered viral life cycle has been postulated [58, 59]. While there is no experimental evidence to fully support this theory, co-emergence of surface glycoprotein mutations together with RdRp mutation also has been reported to occur in a remdesivir-resistant variant of SARS-CoV-2 [60], highlighting the possible role of infection synchronicity (life cycle adaptability) on the potency of antiviral drugs. Nevertheless, in the absence of a reverse genetics system within our current capacity, we were unable to confirm whether the altered life cycle of JUNV imposed by the GP1-A168T substitution plays a direct role in reducing susceptibility to favipiravir.

Here, we experimentally demonstrated that the potency of favipiravir could be significantly enhanced against JUNV if used in combination with ribavirin or remdesivir (**Fig 8**). Furthermore, we showed that it was difficult to isolate JUNV variants that were resistant to the combination treatment (**S4 Fig**). These findings suggest the potential of combination therapies for favipiravir with ribavirin or remdesivir.

In conclusion, we described the isolation of a high replication fidelity variant of arenavirus with reduced susceptibility to favipiravir. More importantly, we provide experimental evidence that hyper-mutagenesis is the primary mechanism of favipiravir action against JUNV. Consistent with our observations, studies on favipiravir treatment of non-human primates infected with Lassa virus showed a reduction in virus infectivity without affecting viral load, providing evidence that favipiravir is primarily a mutagen against old world arenaviruses [61, 62]. Our findings emphasize the importance of the addition of a non-mutagenic inhibitor to the treatment regimens for the AHF.

Furthermore, the antivirals used in this study, namely, favipiravir, ribavirin, and remdesivir, have been reported to have a broad spectrum of antiviral activity. Therefore, combination therapies of these drugs are expected to have a potential therapeutic effect for not only AHF but also the diseases caused by a variety of viruses, including emerging RNA viruses, as the antiviral regimen significantly decreases the possibility of drug-resistant mutant appearance.

## Materials and methods

### Cells, viruses, and compounds

Human embryonic kidney (293T) and African green monkey kidney (Vero 76) cell lines were maintained in Dulbecco's modified Eagle's medium (DMEM; Invitrogen, CA, USA) with 10% fetal bovine serum (FBS) and 1% penicillin and streptomycin. The Candid #1 vaccine strain of JUNV was kindly provided by Dr. Juan C. de la Torre (Scripps Research Institute, California, USA) [63]. Favipiravir was obtained from FUJIFILM Toyama Chemical CO., LTD. (Toyama, Japan). Ribavirin (Sigma Aldrich, MO, USA) and remdesivir (Cayman, MI, USA) were purchased. All compounds were dissolved in 100% dimethyl sulfoxide (DMSO) and stored at −30˚C until use.

### Virus infection and titration

Cells were infected with Candid #1 strain at the indicated multiplicity of infection (MOI). After adsorption for 1 h at 37˚C, the inoculum was removed and washed with PBS (−). Pre-warmed DMEM containing 10% FBS was added to the cells, which were then incubated at 37˚C with 5% $CO_2$. For the quantification of viral titers, a plaque assay was performed according to standard procedures using 10-fold dilutions of the samples in Vero 76 cells as previously described [64].

### Determination of inhibitory concentrations and toxicity testing

To determine the half-maximal inhibitory concentration ($IC_{50}$), 293T cells were infected at a MOI of 0.1 in 24-well plates as explained above. After adsorption, the virus solutions were removed, and fresh DMEM containing serial dilutions of compounds (ranging from 2 μM to 64 μM for favipiravir/ribavirin and 0.0125 μM to 4 μM for remdesivir) were added to the infected cells. At 48 hpi, the supernatants were collected to determine viral titers by plaque assay. To plot the dose-response curve, viral titers from each drug concentration were normalized to the titers in the DMSO control. The cytotoxicity of the compounds was assessed using the CellTiter-Glo cell viability assay (Promega, Madison, WI, USA), following the manufacturer's instructions. Briefly, 293T cells were seeded in a 96-well plate and incubated overnight. Cells were then treated with different concentrations of each compound, as described above. After 48 h, CellTiter-Glo reagent was added, and luminescence was measured using an illuminometer (Tristar LB941, BERTHOLD). Cell viability in DMSO-treated controls was set to 100%.

### Selection and purification of JUNV favipiravir-resistant mutants

To isolate favipiravir-resistant JUNV, we serially passaged the Candid #1 strain in 293T cells at an MOI of 0.01 under the selective pressure of favipiravir (5 μM for the first three passages and 20 μM for the remaining passages). As a control, viruses were serially passaged in the absence of favipiravir in parallel. Supernatants were diluted 10 times in Opti-MEM (Invitrogen) before infecting the cells for the next passages. At 48 hpi, two aliquots of the supernatants were prepared and stored at −80˚C. Virus titers were measured using plaque assays. To isolate a single

clone of the virus, a plaque assay was performed in 6-well plates in Vero 76 cells as described above. After 7 days of incubation, plaques were collected and inoculated into 293T cells to expand the virus clone. To isolate resistant mutants against the combination of favipiravir (0.3 μM) and ribavirin (0.3 μM) or remdesivir (1 nM), the virus was passaged and titrated under similar conditions as stated above.

## Reverse transcription polymerase chain reaction (RT-PCR) and RNA sequencing

RNA was extracted from the supernatant of cells infected with Candid #1 (P0 and P11) using the QIAamp Viral RNA Mini Kit (Qiagen, Hilden, Germany), according to the manufacturer's instructions. For the sequencing, 15 sets of primers were designed to produce overlapping PCR products of 800 to 900 bp (S1 Table) using the Primal Scheme (available at http://primal.zibraproject.org/) [65]. The reference sequences used to design the primers were obtained from the Candid #1 vaccine strain (accession number: AY746354.1 for the L segment and AY746353.1 for the S segment). Viral RNA was amplified using PrimeScript II High Fidelity One Step RT-PCR Kit (Takara Bio, Shiga, Japan) under the following reaction conditions: 45˚C for 10 min, 94˚C for 2 min, 98˚C for 10 s, 55˚C for 15 s, and 68˚C for 10 s, for a total of 30 cycles. The products were then gel-purified using the QIAquick Gel Extraction Kit (Qiagen), according to the manufacturer's instructions. Purified PCR products were sequenced using the BigDye Terminator v3.1 cycle sequencing kit (Thermo Fisher Scientific, MA, USA) and an ABI3500 sequencer (Thermo Fisher Scientific). Consensus sequences were generated and analyzed using GENETYX (GENETYX Corp., Tokyo, Japan) and SnapGene softwares (GSL Biotech; available at snapgene.com). The sequences were submitted to the DNA Data Bank of Japan (DDBJ) (accession numbers: LC637306 for Candid1-P0-RdRp, LC637307 for Candid1-P0-Z, LC637308 for Candid1-P0-GPC, LC637309 for Candid1-P0-NP, LC637310 for Candid1-P11-RdRp, LC637311 for Candid1-P11-Z, LC637312 for Candid1-P11-GPC, and LC637313 for Candid1-P11-NP genes).

## Determination of the proportion of mutations using a next-generation sequencer

RNA was extracted from the supernatant of cells infected with Candid #1 (P2, P4, P6, P8, and P10) using the QIAamp Viral RNA Mini Kit (Qiagen), according to the manufacturer's instructions. The regions including Favipiravir-resistant mutations in the L and S segment were amplified by RT-PCR as described above. After purification using agarose gel electrophoresis, libraries of PCR products were prepared with the NEBNext Ultra II FS DNA Library Prep Kit (New England Biolabs, Ipswich, MA, USA). Sequence analysis was performed using the MiniSeq instrument (Illumina, San Diego, CA, USA) with the MiniSeq High Output Reagent Kit (Illumina). The obtained reads were trimmed and mapped to the reference JUNV Candid #1 sequence using the CLC Genomics Workbench software (Qiagen). The proportion of mutations at each time point was calculated using the number of mapped reads.

## Virus growth analysis

To compare growth kinetics of Candid #1 and Candid #1-res viruses, 293T cells were infected with each virus at an MOI of 0.1 in 24-well plates in duplicates. Infected cells were incubated on ice for 30 min with shaking the plate every 10 min. After inoculum removal, cells were washed twice with DMEM, and fresh media was added. Cells were incubated at 37˚C, and viral titers were measured at 8, 12, 24, and 28 h post infection.

## Determination of the mutation frequency

Candid #1 and Candid #1-res viruses were used to infect 293T cells (MOI: 0.01) in the presence of 20 μM favipiravir or DMSO. At 48 hpi, RNA was extracted from culture supernatants and used to amplify a part of the NP gene using primer number five described in **S1 Table**, with the high-fidelity One Step RT-PCR Kit (Takara Bio). PCR products were gel purified and cloned into the pCR4-TOPO vector using the Zero Blunt Topo Cloning Kit (Invitrogen). The clones were sequenced as described above. A fragment of 450 bp was used for nucleotide polymorphism analysis.

## Nucleoside supplementation assay

293T cells were infected with JUNV (MOI: 0.01). After adsorption at 37°C for 1 h and removal of the inoculum, serial dilutions of the nucleosides adenosine (Sigma), guanosine (Sigma), thymine (Sigma), cytosine (Sigma), and uracil (Sigma) were added to the cells in combination with 50 μM (approximately 10 times the $IC_{50}$) of favipiravir in triplicate. Cells treated with DMSO or favipiravir alone were used as controls. At 48 hpi, a plaque assay was performed to measure viral titers. Cells were visually inspected for any signs of cytotoxicity upon nucleoside treatment, and no toxic effects were observed. Results were expressed as a percentage reduction of the favipiravir anti-JUNV activity.

## Pseudotyped VSV production and virus entry assay

Full-length coding regions of JUNV GPC-A168 (from P0) and GPC-T168 (from P11) were cloned into the pCAGGS mammalian expression vector. Plasmids were designated as pC-GPC-A168 and pC-GPC-T168. Pseudotyped vesicular stomatitis virus (VSV) with a luciferase reporter gene, bearing JUNV GPC, was generated and titrated, as previously described [66,67]. Briefly, 293T cells were seeded in 6-well plates. After 8 h, cells were transfected with 3 μg of either each GPC expression plasmid or pCAGGS empty vector using TransIT LT-1 reagent (Mirus, Madison, WI, USA), according to the manufacturer's instructions. At 24 hpt, cells were infected with G-complemented VSVΔG/Luc and incubated for 1 h at 37°C for adsorption. The cells were washed three times with PBS, and DMEM containing 10% FBS was added to them. Pseudotyped viruses were collected at 24 hpi and labelled as Candid#1pv-A168 and Candid#1pv-T168, respectively. Viruses were stored at −80°C until use. For the internalization assay, a confluent monolayer of 293T cells in a bottom-clear 96-well plate was cooled at 4°C for 10 min and subsequently infected with either Candid#1pv-A168 or Candid#1pv-T168. The plates were further incubated at 4°C for 30 min to allow the binding of viral particles to the receptor without initiation of the entry step [68]. The cells were then washed three times with PBS to remove unbound viral particles. The plates were subsequently incubated at 37°C. Luciferase activity was measured using the Steady-Glo Luciferase Assay System (Promega) and a TriStar LB 941 microplate reader (Berthord Japan K.K., Tokyo, Japan). Since there was a plateau effect at 20 hpi, we considered the signal activity at this time point to be 100%. Similarly, to compare virus binding/attachment efficiencies, a confluent monolayer of 293T cells in 6-well was infected with each virus and allowed to attach at 4°C for 30 min. Virus was then removed, and monolayer was extensively washed to remove un-attached particles. Viral particles were quantified using qPCR as described below.

## Western blotting

Supernatants containing pseudotyped virus were briefly cleared from debris by centrifugation. Ultracentrifugation was performed over a 20% sucrose cushion to pellet virion (60,000 rpm for

30 min at 4˚C). For the detection of intracellular proteins, cells were lysed using lysis buffer (1% NP-40, 50 mM Tris-HCl [pH 8.0], 62.5 mM EDTA, and 0.4% sodium deoxycholate). Prepared samples were analyzed by separation on either 12% (for VSV samples and actin) or 7.5% (for RdRp) sodium dodecyl sulphate–polyacrylamide gels through electrophoresis (SDS-PAGE) and western blotting (WB), as previously described [64]. FLAG-tagged proteins, VSV M protein, or β-Actin were detected using mouse monoclonal primary antibodies against FLAG (M2, F1804, Sigma), VSV M (Kerafast, MA, USA), or β-actin (Sigma), respectively, and HRP-conjugated anti-mouse IgG secondary antibody (Sigma). The labelled proteins were then visualized using ECL prime (GE Healthcare) and LAS3000 (GE Healthcare), according to the manufacturer's instructions. The results were quantified using Multi Gauge software (Fujifilm, Tokyo, Japan).

## Quantitative real time-polymerase chain reaction (qPCR)

Absolute or relative quantification of pseudotyped VSV was performed with a qPCR assay using forward (5′-TGTACATCGGAATGGCAGGG-3′) and reverse (5′-TGCCTTCACAGT GAGCATGATAC-3′) primers specific to the VSV M gene. One-Step TB Green PrimeScript PLUS RT-PCR Kit (Takara Bio) was used under the following conditions: 42˚C for 5 min, 95˚C for 5 s, and 60˚C for 34 s, for a total of 40 cycles using an ABI 7500 thermocycler (Applied Biosystems, Foster City, CA, USA). Plasmids expressing VSV M gene were used as standard. To quantify the encapsidated viral RNA copy numbers, the free RNA not associated with virions was removed from the samples using the Benzonase nuclease (Sigma), according to the manufacturer's instructions, prior to viral RNA extraction. Standard RNA was synthesized from a partial region of the GPC gene using the forward (5′-TAATACGACTCACTATAGGG CCAACCTTTTTGCAGGAGGC-3′) and reverse (5′-AGCTTCTTCTGTGCAGGATCTTCC TGCAAGCGCTAGGAAT-3′) primers and the T7 RNA polymerase (Promega), as previously described [69]. The prepared RNA was then serially diluted using DEPC-treated water to obtain a standard curve ranging from $10^2$–$10^{13}$ copies/mL. To quantify the nano-luciferase mRNA extracted from the minigenome assay, relative qPCR was performed using *GAPDH* expression as a control, as previously described [64], and specific primers targeting the nano-luciferase transcript (Forward, 5′-GGGAGGTGTGTCCAGTTTGT-3′ and reverse, 5′-CCG CTCAGACCTTCATACGG-3′).

## Minigenome assay

To compare the polymerase activities of RdRp-N462 and RdRp-D462, a MG system was constructed based on the Candid #1 S segment. First, the coding region of Candid #1 RdRp was amplified using RNA extracted from P0 or P11 viruses and the PrimeScript II High Fidelity One Step RT-PCR Kit (Takara Bio). Kozak sequence, a FLAG tag (N-terminal), and a linker sequence (5′-GGTAGCGGCAGCGGTAGC-3′) were added through three additional PCR reactions using PrimeStar GXL DNA polymerase (Takara Bio). PCR products were gel-purified in each step, as described elsewhere. The entire fragment was then infused into a pCAGGS expression vector using Infusion HD cloning kit (Takara Bio), according to the manufacturer's instructions. The plasmid expressing JUNV NP, pC-Candid-NP, was kindly provided by Dr Juan C. de la Torre (Scripps Research Institute) [63]. To construct the MG plasmid, sequences of the untranslated regions (UTRs) of the Candid #1-S segment (based on the accession number AY746353) containing the 3′ UTR, 5′ UTR, and intergenic region in an antisense orientation were synthesized (GENEWIZ, NJ, USA). An additional G residue was added upstream of the 3′ UTR to enhance the efficiency of the system [70]. The synthesized fragment was then cloned into a pHH21 plasmid under the control of the human polymerase-I promoter [71].

The nano-luciferase (nluc) reporter gene was then inserted into the NP locus. Further details of the constructs can be provided upon request. To perform the assay, plasmids of Candid #1 NP, MG (with nluc reporter), and RdRp-N462 or RdRp-D462 were transfected into 293T cells at a 1:1:1 ratio using TransIT LT-1 (Mirus, Madison, WI). To normalize transfection efficiency, the pGL4.75 Renilla luciferase (Rluc) plasmid (Promega) was co-transfected. After 24 or 48 h, the cells were lysed and divided into two clear-bottom 96-well plates. Equal volumes of nano-Glo or Renilla-Glo (Promega) were added to measure nluc and Rluc independently. Polymerase activity was determined by the ratio of nluc/Rluc and expressed as relative luciferase induction.

## Drug combination assay and synergy analysis

To evaluate the combinational efficacy of favipiravir with either ribavirin or remdesivir, 293T cells infected with JUNV (MOI: 0.1) were treated with two-fold serially diluted combinations of the drugs at the indicated concentrations. Viral titers were determined at 48 hpi by plaque assay and represented as the percentage inhibition compared to DMSO control for each drug combination. Synergistic inhibition against JUNV growth was determined using SynergyFinder (https://synergyfinder.fimm.fi/) with the Zero Interaction Potency (ZIP) model as previously described [72]. A synergy score (δ-score) of less than −10 is considered as antagonistic, the score range of −10 to 10 suggests an additive drug interaction, and a score greater than 10 indicates a synergistic effect [73].

## Statistical analysis

Non-linear regression analysis was performed to analyze the dose-response of antivirals. The Mann–Whitney $U$ rank test was used to compare the mutational frequency of viruses. Other statistical tests are mentioned in the respective figure legends. All analyses were performed using Prism version 8 (GraphPad Software Inc., La Jolla, CA, USA). Graphical representations were created using the web-based software, BioRender (https://biorender.com/).

## Supporting information

**S1 Table. Primer list used for sequencing of JUNV genome L and S segments.**
(TIF)

**S1 Fig. Determination of favipiravir IC50 value.** 293T cells were infected with Candid #1 (MOI: 0.1). After adsorption, media containing serial dilutions of favipiravir was added. At 48 hpi, supernatant was collected, and viral titers were determined by plaque assay. Error bars indicate ±SD; three independent experiments in duplicates (n = 6) were performed; nonlinear regression analysis was applied.
(TIF)

**S2 Fig. Nucleoside supplementation assay.** 293T cells infected with JUNV (MOI: 0.01) were treated with serial dilutions of nucleosides adenosine, guanosine, thymine, cytosine, and uracil in combination with 50 μM of favipiravir. At 48 hpi, viral titers were measured by plaque assay. Titers were normalized to anti-JUNV activity of favipiravir to estimate the reversal imposed by nucleotide supplementations. Error bars indicate ±SD; two independent experiments in three biological replicates (n = 6) were performed. Statistical significance was determined by 2-way ANOVA tests (ns indicates not significant and *** indicates $P < 0.001$).
(TIF)

**S3 Fig. The amplification results of qPCR assay.** (A) The amplification plot for VSV-M detection from Candid #1pv-A168 (red) or Candid #1pv-T168 (blue) showing a CT value of 27.99 and 27.95 are shown, respectively. (B) Melting curve analysis at the end of the amplification run confirms the specificity of the assay.
(TIF)

**S4 Fig. Serial passaging of JUNV treated with combination of favipiravir and ribavirin or remdesivir.** 293T cells were infected with JUNV at MOI of 0.01 for initial inoculation and 10-fold dilutions for the remaining passages (n = 3). After adsorption, cells were treated with combinations of favipiravir (0.3 μM-16.33 times lower than its $IC_{50}$ value), ribavirin (0.3 μM-20.6 times lower than its $IC_{50}$ value), and remdesivir (1 nM-240 times lower than its $IC_{50}$ value). These concentrations were decided based on synergy assay data. Virus was then titrated at 48 hpi by plaque assay.
(TIF)

**S5 Fig. JUNV Candid #1-mutant virus remains susceptible to ribavirin and remdesivir.** 293T cells were infected with JUNV Candid #1 or Candid #1-mutant virus (MOI: 0.1). Media containing the indicated concentrations of ribavirin was added. At 48 hpi, viral titers were measured by plaque assay. Cytotoxicity assay was performed as described in materials and methods. Error bars indicate ±SD; three independent experiments in duplicate (n = 6) were performed; nonlinear regression analysis was applied.
(TIF)

## Acknowledgments

We are grateful to Drs. Yasuteru Sakurai and Rokusuke Yoshikawa for their support and helpful comments during this research work. We thank Dr. Juan C. de la Torre (Scripps Research Institute, California, USA) for providing the JUNV Candid #1 strain. We also thank Editage (www.editage.com) for English language editing. Favipiravir was kindly provided by FUJI-FILM Toyama Chemical CO., LTD.

## Author Contributions

**Conceptualization:** Jiro Yasuda.

**Data curation:** Vahid Rajabali Zadeh, Haruka Abe, Shuzo Urata, Jiro Yasuda.

**Formal analysis:** Vahid Rajabali Zadeh, Tosin Oladipo Afowowe, Haruka Abe.

**Funding acquisition:** Jiro Yasuda.

**Investigation:** Vahid Rajabali Zadeh, Tosin Oladipo Afowowe, Haruka Abe, Jiro Yasuda.

**Methodology:** Vahid Rajabali Zadeh, Tosin Oladipo Afowowe, Haruka Abe, Shuzo Urata, Jiro Yasuda.

**Project administration:** Jiro Yasuda.

**Supervision:** Shuzo Urata, Jiro Yasuda.

**Validation:** Vahid Rajabali Zadeh, Shuzo Urata, Jiro Yasuda.

**Visualization:** Vahid Rajabali Zadeh.

**Writing – original draft:** Vahid Rajabali Zadeh, Haruka Abe.

**Writing – review & editing:** Jiro Yasuda.

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
