## [Decision Letter · Decision Letter 0]

7 Feb 2022

Dear Prof. Yasuda,

Thank you very much for submitting your manuscript "Potential and action mechanism of favipiravir as an antiviral against Junin virus" for consideration at PLOS Pathogens. As with all papers reviewed by the journal, your manuscript was reviewed by members of the editorial board and by several independent reviewers. In light of the reviews (below this email), we would like to invite the resubmission of a significantly-revised version that takes into account the reviewers' comments.

There are several issues that preclude acceptance of this manuscript in its current form, including placing the work in appropriate context of the field.

We cannot make any decision about publication until we have seen the revised manuscript and your response to the reviewers' comments. Your revised manuscript is also likely to be sent to reviewers for further evaluation.

Sincerely,

Susan R. Ross, PhD

Section Editor

PLOS Pathogens

Susan Ross

Section Editor

PLOS Pathogens

Kasturi Haldar

Editor-in-Chief

PLOS Pathogens

orcid.org/0000-0001-5065-158X

Michael Malim

Editor-in-Chief

PLOS Pathogens

orcid.org/0000-0002-7699-2064

There are several issues that preclude acceptance of this manuscript in its current form, including placing the work in appropriate context of the field.

Reviewer's Responses to Questions

**Part I - Summary**

Reviewer #1: The manuscript covers ground that has been previously examined by a number of groups, many of which are referenced. Previous studies have shown that favipiravir acts as a purine analog (antagonized by authentic purine bases) that introduces mutations and, among other modes of action, pushes the virus to lethal mutagenesis. While the authors conclude that favipiravir resistance can be mediated through increased RdRP fidelity, the evidence for such, and the effect of the N462D mutation itself, is weak (see below). Additionally, others have reported the favipiravir will synergize with ribavirin, another purine analog whose mechanism of action is also complex. Prior studies have shown that ribavirin induces transition mutations, and the authors show the same for favipiravir. Indeed, transitions represent predominant mutations in various RNA viruses, even in the absence of base analogs.

The authors description of the A168T mutation in the virus glycoprotein that arises in favipiravir-resistant virus suggests that they are unaware of the significant literature on this mutation. T168A is one of the important attenuating mutations that arose upon selection of the attenuated Candid#1 vaccine strain of JUNV. The mutation removes a glycosylation site and extensive studies from the Paessler lab suggest that the lack of glycosylation results in GPC instability, reduced GPC levels and induction of the unfolded protein response. The Nunberg lab has shown that this mutation in Candid#1 readily reverts back to 168T upon passage in cell culture as the virus is stressed, either by culture conditions or by unrelated mutation (or by favipiravir?). It is possible that by making the virus more fit, the A168T reversion event can contribute to overcoming the barrier imposed by favipiravir. As the effective load of the more-fit virus is greater, more virions will survive an antiviral effect that renders 90% of the virus population noninfectious. An analogous situation – where fitness is enhanced in a manner that overcomes the initial (albeit unrelated) defect - is described by Chen et al PPath 17(12) e1010191. The authors’ reported decrease in specific infectivity in the resistant virus may thus be the result of the N462D mutation in RdRP, as A168T is expected and indeed reported by the authors to increase infectivity (Fig 6).

Although increased fidelity has been shown to mitigate the effects of base analogs in other systems, the authors’ evidence in this case is weak. The reported reduction in base substitutions in the resistant mutant vs wild-type in the presence of favipiravir is not obviously greater than that reported in the presence of DMSO (21 and 5 versus 5 and 2, respectively), although the absolute numbers are greater in the wild-type than in the mutant (21 and 5, and 5 and 2). Firstly, these numbers are small and conclusions would be strengthened by using global RNA-Seq analysis rather than Sanger sequencing of some 50 individual 450-base clones of NP. (If NP were chosen for sequencing because it does not appear to be a target for favipiravir resistance and thus represents an evolutionarily neutral region, the authors might state their reasoning). More importantly, the reduction in base substitutions may reflect the inhibitory effect of favipiravir treatment on virus replication (greater effect in the wild-type than the resistant mutant, and of course absent in control cultures) and the reduced specific infectivity of the mutant, which would combine to diminish the amount of infectious progeny at each round of virus replication and thereby the amount of RNA replication and the opportunity for mutation. Additional studies are needed to clarify the significance of both mutations. It would be interesting to see if N462D arises upon passage of recombinant A168T Candid#1, or if the A168T virus is itself less sensitive to favipiravir. If the mutation were to increase fidelity, then we might expect to see stronger antagonism by authentic purine bases (see Fig S2) but this study was not performed. If RdRP fidelity is increased, it does not appear to extend to ribavirin or remdesivir as the favipiravir-resistant virus remains sensitive to these (Fig. S5), suggesting that N462D, if indeed responsible for enhanced fidelity, may be specific to favipiravir. Taken together, it is difficult to assert that RdRP fidelity is increased or that N462D is responsible for this or favipiravir resistance.

Specific scientific and editorial comments follow, in page order.

1. line 58-59: might re-order to read “...disease that is endemic in Argentina and caused by...”

2. line 61: rather than the “few reports” on the use of immune plasma therapy, the authors should know that this therapy is used routinely in Argentina.

3. line 83: might replace “antiviral” with the more specific “favipiravir”, and “high” with “strong”.

4. line 93: might replace “invalidates” with “abolishes”.

5. line 107-109: please clarify – as Candid#1 vaccination is quite effective, why would one need to consider favipiravir therapy?

6. para starting line 112: might introduce and provide background information on Candid#1.

7. line 126: this is the first and only mention of “defective interfering particles”, a term that has specific meaning in virology (and in some arenavirus work). If the authors mean to refer to non-infectious genomes containing favipiravir-induced mutations, they might stick with the this nomenclature to avoid confusion with DIPs. Nowhere is “interference” with infection by noninfectious particles discussed.

8. line 146: by “same mutations...as P0 parental virus”, are the authors referring to Candid#1-specific changes relative to pathogenic JUNV (including A168)? One wouldn’t expect too many other changes in a P0 stock. Please clarify.

9. Fig 3: it is remarkable that the authors find extensive virus production at 8 hr post-infection (~3.5 log10 vs ~4.1 log10 at 24 hr). This is a considerably shorter eclipse period than commonly reported by others.

10. line 179: please clarify or cite how the “tolerable threshold of error catastrophe” is “estimated”.

11. line 192: the error bias is observed in the virus population and not the “transitional mutations”. Do the authors refer to “transition” substitutions, mutation that arise transiently during passage, or mutants (vs mutations) that arise on passage? Please clarify.

12. line 208-210: It is likely that multiple rounds of infection take place in 48 hr after a low moi infection, and thus it likely that the lack of difference in final RNA copy number reflects more the duration of each cycle (allowing maximal spread of the more sensitive virus to achieve the same concentration as the less sensitive one) than to any differential inhibition. Please clarify how RNA copy number is the same despite differential sensitivity to favipiravir.

13. line 223, Fig 6A: it is unclear how much weight to place on the one point showing a difference (16 hr) when the two other points (8 and 24 hr) show no difference. Thus, the notion that entry is more rapid in the 168T pseudovirus is questionable.

14. line 243: might specify that “no...effect” is in the absence of favipiravir.

15. line 249-251 (and line 291): please explain how Fig 7F speaks to chain termination?

16. line 267-269: might reword that it’s the gradual increase in selective pressure (favipiravir concentration; not “mutant viruses”) that enables adequate levels of replication to support mutation and the emergence of resistant viruses. This is a typical design for selection.

17. line 293: the authors claim ribavirin is non-mutagenic, yet literature claims otherwise? Please clarify.

18. line 321-323, and Fig S4: the failure to identify resistance to the combination of favipiravir and ribavirin/remdesivir is rather unsurprising – at a minimum, one would need 2 (vs 1) mutation, and it is unclear whether the effective (synergistic) potency was considered in establishing selection conditions – selection may fail if replication is excessively reduced. However, Fig S4 appears to show no antiviral effect of the combination, as DMSO and combination tracks overlap (compare with Fig1A, where treatment with favipiravir alone does reduce virus output). This too would explain the absence of selection. Please explain.

Reviewer #2: In “Potential and action mechanism of favipiravir as an antiviral against Junin virus” Zadeh et al. describe the generation of favipiravir-resistant JUNV mutants, and further investigate the potential mechanisms of resistance. The objectives of the study are well defined, with a strong experimental method, clear figures, and well written results and discussion sections. The data presented is sound, and represents a thorough investigation into the resistance mechanisms conferred by the two mutations identified.

The main comments I have on the data that would strengthen the manuscript are that: 1) relative appearance of the two identified mutations; 2) some data that has been generated is not included in this manuscript (data not shown); and 3) no data investigating these mutations in the more clinically relevant virulent JUNV strains.

**Part II – Major Issues: Key Experiments Required for Acceptance**

Reviewer #1: The authors description of the A168T mutation in the virus glycoprotein that arises in favipiravir-resistant virus suggests that they are unaware of the significant literature on this mutation. T168A is one of the important attenuating mutations that arose upon selection of the attenuated Candid#1 vaccine strain of JUNV. The mutation removes a glycosylation site and extensive studies from the Paessler lab suggest that the lack of glycosylation results in GPC instability, reduced GPC levels and induction of the unfolded protein response. The Nunberg lab has shown that this mutation in Candid#1 readily reverts back to 168T upon passage in cell culture as the virus is stressed, either by culture conditions or by unrelated mutation (or by favipiravir?). It is possible that by making the virus more fit, the A168T reversion event can contribute to overcoming the barrier imposed by favipiravir. As the effective load of the more-fit virus is greater, more virions will survive an antiviral effect that renders 90% of the virus population noninfectious. An analogous situation – where fitness is enhanced in a manner that overcomes the initial (albeit unrelated) defect - is described by Chen et al PPath 17(12) e1010191. The authors’ reported decrease in specific infectivity in the resistant virus may thus be the result of the N462D mutation in RdRP, as A168T is expected and indeed reported by the authors to increase infectivity (Fig 6).

Although increased fidelity has been shown to mitigate the effects of base analogs in other systems, the authors’ evidence in this case is weak. The reported reduction in base substitutions in the resistant mutant vs wild-type in the presence of favipiravir is not obviously greater than that reported in the presence of DMSO (21 and 5 versus 5 and 2, respectively), although the absolute numbers are greater in the wild-type than in the mutant (21 and 5, and 5 and 2). Firstly, these numbers are small and conclusions would be strengthened by using global RNA-Seq analysis rather than Sanger sequencing of some 50 individual 450-base clones of NP. (If NP were chosen for sequencing because it does not appear to be a target for favipiravir resistance and thus represents an evolutionarily neutral region, the authors might state their reasoning). More importantly, the reduction in base substitutions may reflect the inhibitory effect of favipiravir treatment on virus replication (greater effect in the wild-type than the resistant mutant, and of course absent in control cultures) and the reduced specific infectivity of the mutant, which would combine to diminish the amount of infectious progeny at each round of virus replication and thereby the amount of RNA replication and the opportunity for mutation. Additional studies are needed to clarify the significance of both mutations. It would be interesting to see if N462D arises upon passage of recombinant A168T Candid#1, or if the A168T virus is itself less sensitive to favipiravir. If the mutation were to increase fidelity, then we might expect to see stronger antagonism by authentic purine bases (see Fig S2) but this study was not performed. If RdRP fidelity is increased, it does not appear to extend to ribavirin or remdesivir as the favipiravir-resistant virus remains sensitive to these (Fig. S5), suggesting that N462D, if indeed responsible for enhanced fidelity, may be specific to favipiravir. Taken together, it is difficult to assert that RdRP fidelity is increased or that N462D is responsible for this or favipiravir resistance.

Reviewer #2: Firstly, in figure 1 it is apparent that between P10 and p11 the resistant mutation (with the 2 mutations) was generated. However, the authors only present sequence data from the P0 and P11 strains. I think the manuscript would be strengthen if the viruses at some of the intermediate passage numbers were also sequenced, and therefore the relative appearance of each individual mutation could be elucidated. From a readers point of view, I would be interested in knowing (especially given the later data examining the relative importance of each mutation) whether one came before the other, or did both appear at P11 together? This would also generate data which could strengthen the conclusions on the discussion, knowing which of the mutations (if any) is the more relevant for the resistance observed here.

Secondly, one comment I had regarding the GPC-A168T mutation was regarding the effect this had on cellular entry, specifically attachment. Although the experiment was performed (with the pseudotype) these data were not included in the manuscript. I think however that, if possible, these data should be included as a panel in figure 6 as I believe, even if it is negative data, it is an important finding. It would also clearly demonstrate the thoroughness to which the authors have investigated the potential mechanisms of the GPC-A168T mutation.

Finally, although the authors state that there is no reverse genetics system for JUNV, there is published data of a system (Albariño et al. Efficient reverse genetics generation of infectious Junin viruses differing in glycoprotein processing. J Virol. 2009 PMID: 19321606). However, I understand that without a BSL4 lab this would be impossible to set up, I wondered if the authors had given any thought to collaborating with a lab that does have this facility to generate a JUNV with these mutations to look at resistance in more clinically relevant strains?

**Part III – Minor Issues: Editorial and Data Presentation Modifications**

Reviewer #1: Specific scientific and editorial comments follow, in page order.

1. line 58-59: might re-order to read “...disease that is endemic in Argentina and caused by...”

2. line 61: rather than the “few reports” on the use of immune plasma therapy, the authors should know that this therapy is used routinely in Argentina.

3. line 83: might replace “antiviral” with the more specific “favipiravir”, and “high” with “strong”.

4. line 93: might replace “invalidates” with “abolishes”.

5. line 107-109: please clarify – as Candid#1 vaccination is quite effective, why would one need to consider favipiravir therapy?

6. para starting line 112: might introduce and provide background information on Candid#1.

7. line 126: this is the first and only mention of “defective interfering particles”, a term that has specific meaning in virology (and in some arenavirus work). If the authors mean to refer to non-infectious genomes containing favipiravir-induced mutations, they might stick with the this nomenclature to avoid confusion with DIPs. Nowhere is “interference” with infection by noninfectious particles discussed.

8. line 146: by “same mutations...as P0 parental virus”, are the authors referring to Candid#1-specific changes relative to pathogenic JUNV (including A168)? One wouldn’t expect too many other changes in a P0 stock. Please clarify.

9. Fig 3: it is remarkable that the authors find extensive virus production at 8 hr post-infection (~3.5 log10 vs ~4.1 log10 at 24 hr). This is a considerably shorter eclipse period than commonly reported by others.

10. line 179: please clarify or cite how the “tolerable threshold of error catastrophe” is “estimated”.

11. line 192: the error bias is observed in the virus population and not the “transitional mutations”. Do the authors refer to “transition” substitutions, mutation that arise transiently during passage, or mutants (vs mutations) that arise on passage? Please clarify.

12. line 208-210: It is likely that multiple rounds of infection take place in 48 hr after a low moi infection, and thus it likely that the lack of difference in final RNA copy number reflects more the duration of each cycle (allowing maximal spread of the more sensitive virus to achieve the same concentration as the less sensitive one) than to any differential inhibition. Please clarify how RNA copy number is the same despite differential sensitivity to favipiravir.

13. line 223, Fig 6A: it is unclear how much weight to place on the one point showing a difference (16 hr) when the two other points (8 and 24 hr) show no difference. Thus, the notion that entry is more rapid in the 168T pseudovirus is questionable.

14. line 243: might specify that “no...effect” is in the absence of favipiravir.

15. line 249-251 (and line 291): please explain how Fig 7F speaks to chain termination?

16. line 267-269: might reword that it’s the gradual increase in selective pressure (favipiravir concentration; not “mutant viruses”) that enables adequate levels of replication to support mutation and the emergence of resistant viruses. This is a typical design for selection.

17. line 293: the authors claim ribavirin is non-mutagenic, yet literature claims otherwise? Please clarify.

18. line 321-323, and Fig S4: the failure to identify resistance to the combination of favipiravir and ribavirin/remdesivir is rather unsurprising – at a minimum, one would need 2 (vs 1) mutation, and it is unclear whether the effective (synergistic) potency was considered in establishing selection conditions – selection may fail if replication is excessively reduced. However, Fig S4 appears to show no antiviral effect of the combination, as DMSO and combination tracks overlap (compare with Fig1A, where treatment with favipiravir alone does reduce virus output). This too would explain the absence of selection. Please explain.

Reviewer #2: Figure 1: I think it would be helpful to indicate on the x-axis where the dose of T705 was increased, or indicate the dose at each passage maybe?

Figure 6A: I think y-axis should say normalized? And indicate % as a unit?

Line 344: Please list Genbank number of Cd#1 strain used (if available)

Line 733: “LOD” not shown on graph but listed in legend.

Line 741: Should say “cells” and not “cell”.

Line 747: LOD is shown on graph but not explained in legend.

Line 761: Unsure how the n is 8 for this?

Line 815: Unsure how the n is 6 for this?

Lie 831: Should say “cells” and not “cell”.

PLOS authors have the option to publish the peer review history of their article (what does this mean?). If published, this will include your full peer review and any attached files.

Reviewer #1: No

Reviewer #2: No
---

## [Decision Letter · Decision Letter 1]

19 Jun 2022

Dear Prof. Yasuda,

We are pleased to inform you that your manuscript 'Potential and action mechanism of favipiravir as an antiviral against Junin virus' has been provisionally accepted for publication in PLOS Pathogens.

Best regards,

Amy L Hartman, PhD

Associate Editor

PLOS Pathogens

Susan Ross

Section Editor

PLOS Pathogens

Kasturi Haldar

Editor-in-Chief

PLOS Pathogens

orcid.org/0000-0001-5065-158X

Michael Malim

Editor-in-Chief

PLOS Pathogens

orcid.org/0000-0002-7699-2064

Reviewer Comments (if any, and for reference):

Reviewer's Responses to Questions

**Part I - Summary**

Reviewer #2: The authors have addressed all of my concerns and incorporated the edits/suggestions.

**Part II – Major Issues: Key Experiments Required for Acceptance**

Reviewer #2: The authors have addressed all of my concerns and incorporated the edits/suggestions.

**Part III – Minor Issues: Editorial and Data Presentation Modifications**

Reviewer #2: The authors have addressed all of my concerns and incorporated the edits/suggestions.

PLOS authors have the option to publish the peer review history of their article (what does this mean?). If published, this will include your full peer review and any attached files.

Reviewer #2: No

---

## [Editor Report · Acceptance letter]

1 Jul 2022

Dear Prof. Yasuda,

We are delighted to inform you that your manuscript, "Potential and action mechanism of favipiravir as an antiviral against Junin virus," has been formally accepted for publication in PLOS Pathogens.

Best regards,

Kasturi Haldar

Editor-in-Chief

PLOS Pathogens

orcid.org/0000-0001-5065-158X

Michael Malim

Editor-in-Chief

PLOS Pathogens

orcid.org/0000-0002-7699-2064